# Distinct chromatin functional states correlate with HIV latency reactivation in infected primary CD4[+] T cells

Emilie Battivelli[1,2,3], Matthew S Dahabieh[1,2], Mohamed Abdel-Mohsen[4,5,6], J Peter Svensson[7], Israel Tojal Da Silva[8,9], Lillian B Cohn[8], Andrea Gramatica[1,2,10], Steven Deeks[2], Warner C Greene[1,2,10], Satish K Pillai[4,5], Eric Verdin[1,2,3]*

[1]Gladstone Institute of Virology and Immunology, Gladstone Institutes, San Francisco, United States; [2]Department of Medicine, University of California San Francisco, San Francisco, United States; [3]Buck Institute for Research on Aging, Novato, United States; [4]University of California San Francisco, San Francisco, United States; [5]Blood Systems Research Institute, San Francisco, United States; [6]The Wistar Institute, Philadelphia, United States; [7]Department of Biosciences and Nutrition, Karolinska Institutet, Solna, Sweden; [8]Laboratory of Molecular Immunology, The Rockefeller University, New York, United States; [9]Laboratory of Computational Biology and Bioinformatics, International Research Center, Sao Paulo, Brazil; [10]Department of Cellular and Molecular Pharmacology, University of California San Francisco, San Francisco, United States

*For correspondence:
EVerdin@buckinstitute.org

Competing interests: The authors declare that no competing interests exist.

**Abstract** Human immunodeficiency virus (HIV) infection is currently incurable, due to the persistence of latently infected cells. The 'shock and kill' approach to a cure proposes to eliminate this reservoir via transcriptional activation of latent proviruses, enabling direct or indirect killing of infected cells. Currently available latency-reversing agents (LRAs) have however proven ineffective. To understand why, we used a novel HIV reporter strain in primary CD4[+] T cells and determined which latently infected cells are reactivatable by current candidate LRAs. Remarkably, none of these agents reactivated more than 5% of cells carrying a latent provirus. Sequencing analysis of reactivatable vs. non-reactivatable populations revealed that the integration sites were distinguishable in terms of chromatin functional states. Our findings challenge the feasibility of 'shock and kill', and suggest the need to explore other strategies to control the latent HIV reservoir.
DOI: https://doi.org/10.7554/eLife.34655.001

## Introduction

Antiretroviral therapy (ART) has transformed HIV infection from a uniformly deadly disease into a chronic lifelong condition, saving millions of lives. However, ART interruption leads to rapid viral rebound within weeks due to the persistence of proviral latency in rare, long-lived resting CD4[+] T cells and possibly in tissue macrophages (*Honeycutt et al., 2017*). HIV latency is defined as the presence of a transcriptionally silent but replication-competent proviral genome. Latency allows infected cells to evade both immune clearance mechanisms and currently available ART, which is based solely on the elimination of actively replicating virus.

An extensively investigated approach to purging latent HIV is the 'shock and kill' strategy, which consists of forcing the reactivation of latent proviruses ('shock' phase) with the use of latency-reversing agents (LRAs), while maintaining ART to prevent de novo infections. Subsequently, reactivation

of HIV expression would expose such cells (shocked cells) to killing by viral cytopathic effects and immune clearance ('kill' phase). A variety of LRAs have been explored in vitro and ex vivo, with only a few candidates being advanced to testing in pilot human clinical trials. Use of histone deacetylase inhibitors (HDACi: vorinostat, panobinostat, romidepsin, and disulfiram) in clinical studies has shown increases in cell-associated HIV RNA production and/or plasma viremia after in vivo administration (*Archin et al., 2012a*; *Elliott et al., 2015*; *Elliott et al., 2014*; *Rasmussen et al., 2014*; *Søgaard et al., 2015*). However, none of these interventions alone has succeeded in significantly reducing the size of the latent HIV reservoir (*Rasmussen and Lewin, 2016*).

Several obstacles can explain the failure of LRAs, as reviewed in (*Margolis et al., 2016*; *Rasmussen et al., 2016*). However, the biggest challenge to date is our inability to accurately quantify the size of the reservoir. The absolute quantification (number of cells) of the latent reservoir in vivo (and ex vivo) has thus far been technically impossible. The most sensitive, quickest, and easiest assays to measure the prevalence of HIV-infected cells are PCR-based, quantifying total or integrated HIV DNA or RNA transcripts. However these assays substantially overestimate the number of latently infected cells, due to the predominance of defective HIV DNA genomes in vivo (*Bruner et al., 2016*; *Ho et al., 2013*). The best currently available assay to measure the latent reservoir is the relatively cumbersome viral outgrowth assay (VOA), which is based on quantification of the number of resting CD4$^+$ T cells that produce infectious virus after a single round of maximum in vitro T-cell activation. After several weeks of culture, viral outgrowth is assessed by an ELISA assay for HIV-1 p24 antigen or a PCR assay for HIV-1 RNA in the culture supernatant. Importantly, the number of latently infected cells detected in the VOA is 300-fold lower than the number of resting CD4$^+$ T cells that harbor proviruses detectable by PCR.

This reliance on a single round of T-cell activation likely incorrectly estimates the viral reservoir for two reasons. First, the discovery of intact non-induced proviruses indicates that the size of the latent reservoir may be much greater than previously thought: the authors estimate that the number may be at least 60 fold higher than estimates based on VOA (*Ho et al., 2013*; *Sanyal et al., 2017*). This work and that of others (*Chen et al., 2017*) highlight the heterogeneous nature of HIV latency and suggest that HIV reactivation is a stochastic process that only reactivates a small fraction of latent viruses at any given time (*Dar et al., 2012*; *Ho et al., 2013*; *Singh et al., 2010*; *Weinberger et al., 2005*). Second, the ability of defective proviruses to be transcribed and translated in vivo (*Pollack et al., 2017*): this study shows that, although defective proviruses cannot produce infectious particles, they express viral RNA and proteins, which can be detectable by any p24 antigen or PCR assays used for the reservoir-size quantification.

Thus, current assays underestimate the actual number of latently infected cells, both in vivo and ex vivo, and the real size of HIV reservoir is still to be determined. Therefore, it has been difficult to judge the potential of LRAs in in vitro (latency primary models), ex-vivo (patients' samples) and in vivo (clinical trial) experiments.

HIV latency is a complex, multi-factorial process (reviewed in [*Dahabieh et al., 2015*]). Its establishment and maintenance depend on: (a) viral factors, such as integrase that specifically interacts with cellular proteins, including LEDGF, (b) *trans*-acting factors (e.g., transcription factors) and their regulation by the activation state of T cells and the environmental cues that these cells receive, and (c) *cis*-acting mechanisms, such as the local chromatin environment at the site of integration of the virus into the genome. Recent evidence has also highlighted the association of specific HIV-1 integration sites with clonal expansion of latently infected cells (reviewed in [*Maldarelli, 2016*]).

The role of the site of HIV integration into the cellular genome in the establishment and maintenance of HIV latency has remained controversial. While early studies found that the HIV integration site does affect both the entry into latency (*Chen et al., 2017*; *Jordan et al., 2003*; *Jordan et al., 2001*), and the viral response to LRAs (*Chen et al., 2017*), other studies have failed to find a significant role of integration sites in regulating the fate of HIV infection (*Dahabieh et al., 2014*; *Sherrill-Mix et al., 2013*).

In this study, we have used a new dual color reporter virus, HIV$_{GKO}$, to investigate the reactivation potential of various LRAs in pure latent population. We find that latency is heterogeneous and that only a small fraction (<5%) of the latently infected cells is reactivated by LRAs. We also show that both genomic localization and chromatin context of the integration site affect the fate of HIV infection and the reversal of viral latency.

## Results

### A second-generation dual-fluorescence HIV-1 reporter (HIV$_{GKO}$) to study latency

Our laboratory reported the development of a dual-labeled virus (DuoFluoI) in which eGFP is under the control of the HIV-1 promoter in the 5′ LTR and mCherry is under the control of the cellular elongation factor one alpha promoter (EF1α) (*Calvanese et al., 2013*). However, we noted that the model was limited by a modest number of latently infected cells (<1%) generated regardless of viral input (*Figure 1—figure supplement 1A–1C*), as well as a high proportion of productively infected cells in which the constitutive promoter EF1α was not active (GFP+, mCherry-).

To address these issues, which we suspected were due to recombination between the 20–30 bp regions of homology at the N- and C-termini of the adjacent fluorescent proteins (eGFP and mCherry) (*Salamango et al., 2013*), we generated a new version of dual-labeled virus (HIV$_{GKO}$), containing a codon-switched eGFP (csGFP) and a distinct, unrelated fluorescent protein mKO2 under the control of EF1α (*Figure 1A*). First, titration of HIV$_{GKO}$ input revealed that productively and latently infected cells increased proportionately as the input virus increased (*Figure 1B and Figure 1—figure supplement 1*), unlike the original DuoFluoI (*Figure 1—figure supplement 1*). Second, comparison of primary CD4$^+$ T cells infected with HIV$_{GKO}$ or the original DuoFluoI revealed an increase in double-positive (csGFP+ mKO2+) infected cells in HIV$_{GKO}$ infected cells (*Figure 1C*). A small proportion of csGFP+ mKO2- cells were still visible in HIV$_{GKO}$ infected cells. We generated a HIV$_{GKO}$ virus lacking the U3 promoter region of the 3′LTR (ΔU3-GKO), resulting in an integrated virus devoid of the 5′ HIV U3 region. This was associated with a suppression of HIV transcription and an inversion of the latency ratio (ratios latent/productive = 0.34 for HIV$_{GKO-WT-LTR}$ and 8.8 for HIV$_{GKO-\Delta U3-3'LTR}$ - *Figure 1D*). Finally, to further characterize the constituent populations of infected cells, double-negative cells, latently and productively infected cells were sorted using FACS and analyzed for viral mRNA and protein content. (*Figures 1E and F*, *Figure 1—source data 1*). As expected, productively infected cells (csGFP+) expressed higher amounts of viral mRNA and viral proteins, but latently infected cells (csGFP- mKO2+) had very small amounts of viral mRNA and no detectable viral proteins.

Based on all these findings, the second-generation of dual-fluorescence reporter, HIV$_{GKO}$, is able to more accurately quantify latent infections in primary CD4$^+$ T cells than HIV$_{DuoFluoI}$, and thus allows for the identification and purification of a larger number of latently infected cells. Using flow cytometry, we can determine infection and HIV productivity of individual cells and simultaneously control for cell viability.

### Correlation between LRA efficacy in HIV-infected patient samples and activity in HIV$_{GKO}$ latently infected cells

Next, we evaluated the reactivation of latent HIV$_{GKO}$ in primary CD4$^+$ T cells by LRAs, and compared it with the ability of the same LRAs to reverse latency in CD4$^+$ T cells isolated from HIV-infected individuals. We tested the following LRAs: (a) the histone deacetylase inhibitor (HDACi) panobinostat (*Rasmussen et al., 2013*), (b) the bromodomain-containing protein 4 (BRD4) inhibitor JQ1, which acts through positive transcription elongation factor (P-TEFb) (*Banerjee et al., 2012*; *Boehm et al., 2013*; *Filippakopoulos et al., 2010*; *Li et al., 2013*; *Zhu et al., 2012*), and (c) the PKC activator, bryostatin-1 (*del Real et al., 2004*; *Mehla et al., 2010*). Viral reactivation mediated by these LRAs was compared to treatment of CD4$^+$ T cells with αCD3/CD28 (*Spina et al., 2013*). Several studies have shown synergetic effects when combining different LRAs (*Darcis et al., 2015*; *Jiang et al., 2015*; *Laird et al., 2015*; *Martínez-Bonet et al., 2015*), therefore we also tested bryostatin-1 in combination with either panobinostat or JQ1. Drugs were used at concentrations previously shown to be effective at reversing latency in other model systems (*Archin et al., 2012a*; *Bullen et al., 2014*; *Laird et al., 2015*; *Spina et al., 2013*).

To measure reactivation by LRAs in patient samples, we treated 5 million purified resting CD4$^+$ T cells from four HIV infected individuals on suppressive ART (participant characteristics in *Table 1*) with single LRAs, combinations thereof, or vehicle alone for 24 hr. LRAs efficacy was assessed using a PCR-based assay, by measuring levels of intracellular HIV-1 RNA using primers and a probe that detect the 3′ sequence common to all correctly terminated HIV-1 mRNAs (*Bullen et al., 2014*). Of

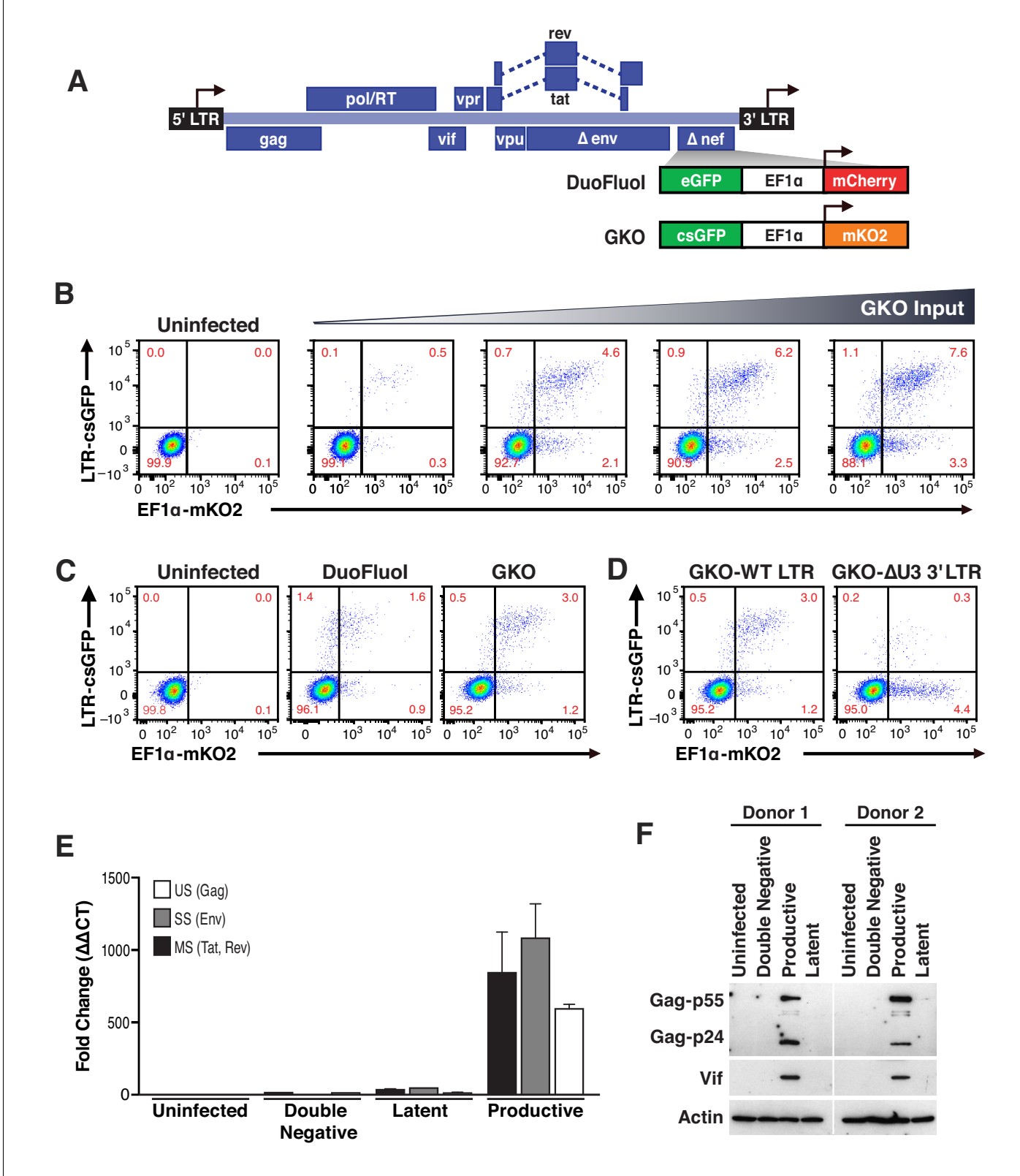

**Figure 1.** Second generation of dual-fluorescence HIV-1 reporter, HIV_GKO to quantify stable latency. (**A**) Schematic representation of first (top: HIV_DuoFluoI) and second generation (bottom: HIV_GKO) of dual-labeled HIV-1 reporters. (**B**) Representative experiment of HIV_GKO virus titration in activated primary CD4[+] T cells (4 days post-infection). Primary CD4[+] T cells were activated with αCD3/CD28 beads + 20 U/mL IL-2 for 3 days before infection with different amounts of HIV_GKO (input, ng/p24) and analyzed by flow cytometry 4 days post-infection. (**C**) Comparison of HIV_DuoFluoI and

*Figure 1 continued on next page*

*Figure 1 continued*

HIV$_{GKO}$ infection profiles by flow cytometry in activated primary CD4$^+$ T-cells (4 days post-infection). Cells were treated as in (B). (D) Comparison of GKO-WT-LTR and GKO-ΔU3 3'LTR infection profiles by flow cytometry in cells treated as in (B). (E, F) Primary CD4$^+$ T cells were treated as in (B). At 4 days post-infection, double-negative, productively infected, and latently infected cells were sorted out, and (E) the total RNA isolated from each population was subjected to Taqman RT-qPCR analysis (Source Data - *Figure 1*). Unspliced (US), singly spliced (SS), and multiply spliced (MS) HIV-1 mRNAs were quantified relative to cellular GAPDH. (F) Western blot analysis of each population.
DOI: https://doi.org/10.7554/eLife.34655.002

The following source data and figure supplement are available for figure 1:

**Source data 1.** Taqman RT-qPCR analysis of unspliced (US), singly spliced (SS), and multiply spliced (MS) HIV-1 mRNAs in the uninfected, double negative, latent and productive populations.
DOI: https://doi.org/10.7554/eLife.34655.004

**Figure supplement 1.** Comparison of HIV$_{GKO}$ and HIV$_{DuoFluoI}$.
DOI: https://doi.org/10.7554/eLife.34655.003

the LRAs tested individually, none exhibited a statistically significant effect (n=4 - *Figure 2A*, *Figure 2—source data 1*). Importantly, T-cell activation positive control, αCD3/CD28 (24.4-fold, *Figure 2A*), showed expected fold induction value (10 to 100-fold increases of HIV RNA in PBMCs [*Bullen et al., 2014*; *Darcis et al., 2015*; *Laird et al., 2015*]). Combinations of the PKC agonist bryostatin-1 with JQ1 or with panobinostat (fold-increases of 126.2- and 320.8-fold, respectively, *Figure 2A*), were highly more effective than bryostatin-1, JQ1 or panobinostat alone (fold-increases of 6.8, 1.7- and 2.9-fold, respectively, *Figure 3A*), and even greater than T-cell activation with αCD3/CD28. This observation is consistent with previous reports (*Darcis et al., 2015*; *Jiang et al., 2015*; *Laird et al., 2015*; *Martínez-Bonet et al., 2015*).

The same LRAs and combinations were next tested after infection of human CD4$^+$T cells in vitro with HIV$_{GKO}$. Measurement of intracellular HIV-1 mRNA in HIV$_{GKO}$ latently infected cells showed an expected fold induction of latency in response to αCD3/CD28 (11.3-fold, *Figure 2B*, *Figure 2—source data 1*). Second, JQ1, panobinostat, and bryostatin-1 alone all caused limited reactivation of latent HIV (fold-increases of 1.1-, 5.6- and 6.2-fold, respectively, *Figure 2B*), as observed in patients' samples. Finally, we observed low synergy when combining bryostatin and JQ1 (8-fold increase), but high synergy between bryostatin and panobinostat (67.3-fold increase). These data together demonstrate that HIV$_{GKO}$ closely mimics in vitro what is observed in ex vivo patients' samples (correlation rate $r^2$ = 0.88, p=0.0056 - *Figure 2C*), and validate the robustness and reliability of the dual-florescence HIV reporter as a model to study HIV-1 latency.

## HIV-1 LRAs target a minority of latently infected primary CD4$^+$ T cells

Current assays have relied on PCR-based assays to measure HIV RNA, and to evaluate the efficacy of different LRAs (*Figure 2A*). The use of dual-fluorescent HIV reporters, however, provides a tool to quantify directly the fraction of cells that become reactivated.

To quantify the absolute number of latently infected cells reactivated following LRA treatment, primary CD4$^+$ T cells were infected with HIV$_{GKO}$, and cultured for 5 days (in the presence of IL-2) before sorting the pure latent population (GFP-, mKO2+). Cells were allowed to rest overnight and

**Table 1.** Characteristics of HIV-1-infected study participants

ABC, abacavir; DRV, darunavir; FTC, emtricitabine; RPV, rilpivirine; RTV, ritonavir; TCV, tivicay; TDF, tenofovir; 3TC, lamivudine; VL, viral load.

| Scope ID | Age | Sex | Ethnicity | CD4 Count | Duration of infection (years) | ART regimen | Duration of ART (years) | Peak reporter VL (copies/ml$^{-1}$) |
|---|---|---|---|---|---|---|---|---|
| 1597 | 56 | M | Mixed | 469 | 19 | RPV/TDF/FTC | 5 | 45734 |
| 2147 | 59 | M | Asian | 597 | 28 | RPV/TDF/FTC | 23 | 374000 |
| 2461 | 62 | M | White | 664 | 32 | RPV/TCV | 19 | 20000 |
| 3162 | 54 | M | White | 734 | 29 | RTV, DRV, ABC/TCV/3TC | 20 | 171000 |

DOI: https://doi.org/10.7554/eLife.34655.009

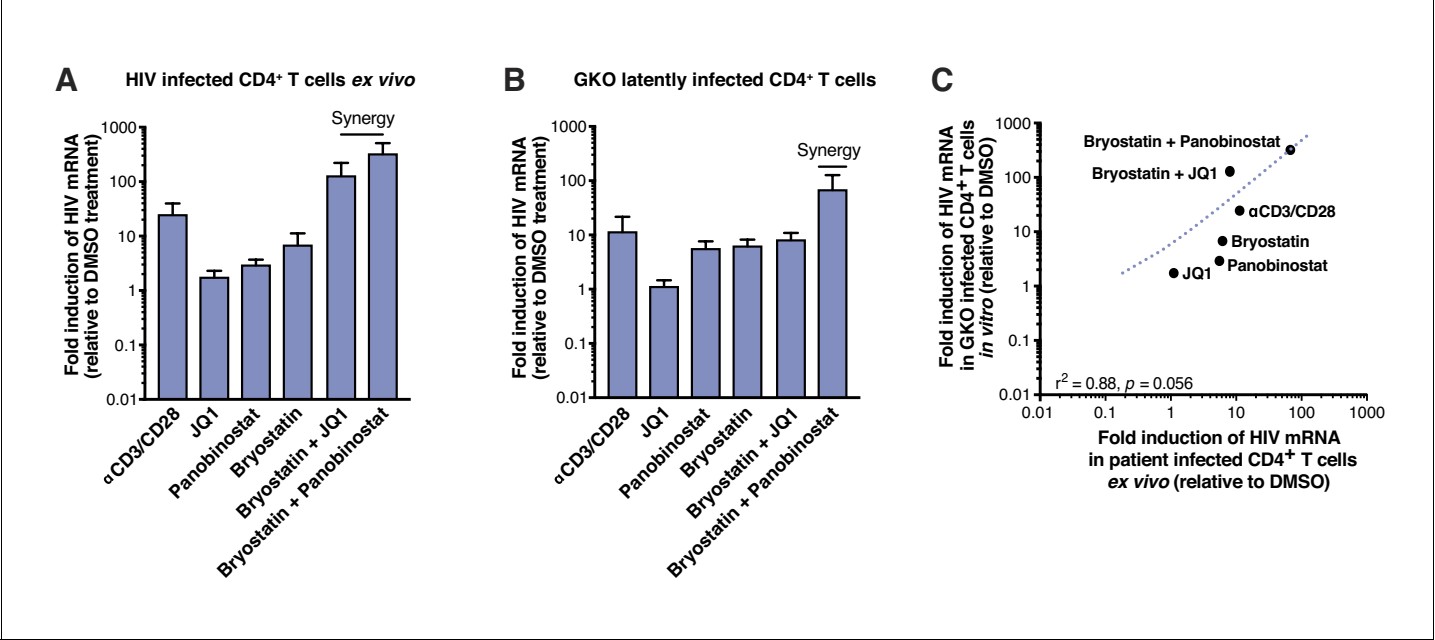

**Figure 2.** LRAs efficacy in patient samples is predicted by activity in HIV_GKO latently infected cells. (**A**) Intracellular HIV-1 mRNA levels in rCD4s, obtained from infected individuals and treated ex vivo with a single LRA or a combination of two LRAs for 24 hr in presence of raltegravir, presented as fold induction relative to DMSO control. (n = 4, mean +SEM) (*Figure 2—source data 1*). (**B**) Intracellular HIV-1 mRNA levels in HIV_GKO latently infected CD4$^+$ T-cells, and treated with a single LRA or a combination of two LRAs for 6 hr in presence of raltegravir, presented as fold induction relative to DMSO control. (n = 3 (different donors), mean +SEM, paired t-test) (*Figure 2—source data 1*). (**C**) Correlation between intracellular HIV-1 mRNA levels quantified in either 6 hr stimulated HIV_GKO latently infected CD4$^+$ T-cells from different donors, or 24 hr stimulated rCD4s from HIV infected patients, with a single LRA or a combination of two LRAs in presence of raltegravir.

DOI: https://doi.org/10.7554/eLife.34655.005

The following source data is available for figure 2:

**Source data 1.** Intracellular HIV-1 mRNA levels in rCD4s, obtained from infected individuals, or in HIV_GKO latently infected CD4$^+$ T-cells.
DOI: https://doi.org/10.7554/eLife.34655.006

were treated for 24 hr with the various LRAs (same drug concentrations as in *Figure 2*) (*Figure 3A*, *Figure 3—source data 1*). Culture of DMSO-treated latently infected primary CD4$^+$ T cells produced little spontaneous reactivation (average of four experiments: 1.4% of GFP+ cells). Unexpectedly, we found that none of the individual LRAs or their combinations reactivated more than 5.5% of the latently infected cells: JQ1 (1.7%) panobinostat (3.7%), bryostatin-1 (3%) αCD3/CD28 (4.5%), bryostatin-1 and JQ1 (3.3%.). bryostatin-1 and panobinostat (5.5%) (*Figure 3B*).

## Small fractional rate of latency reactivation is not explained by low cellular response to activation signals

These data highlight two important facts: a) cell-associated HIV RNA quantification does not reflect the absolute number of cells undergoing viral reactivation, and b) induced cell-associated HIV RNA, in response to all reversing agents, comes from a small fraction of reactivated latent cells. This was particularly surprising with αCD3/CD28 stimulation, as a currently accepted model for HIV latency is that the state of T cell activation dictates the transcriptional state of the provirus. Treatment of latently infected primary CD4$^+$ T cells with αCD3/CD28 stimulated HIV production in less than 5% of the cells, while the other 95% remained latent, even though after 24 hr of treatment nearly all of the cells had upregulated the early T cell activation marker CD69 (*Figure 4—figure supplement 1*).

To further rule out the possibility that non-reactivated latently infected cells (NRLIC) simply represented a lack of efficient response to T-cell activation signals, we analyzed T-cell activation markers within the different populations (i.e., within uninfected, non reactivated (NRLIC) and reactivated latently infected cells (RLIC); *Figure 4*, *Figure 4—source data 1*). Briefly, 72h-stimulated CD4$^+$ T cells were infected with HIV_GKO, and 4 days later, GFP- cells were sorted, and allowed to rest

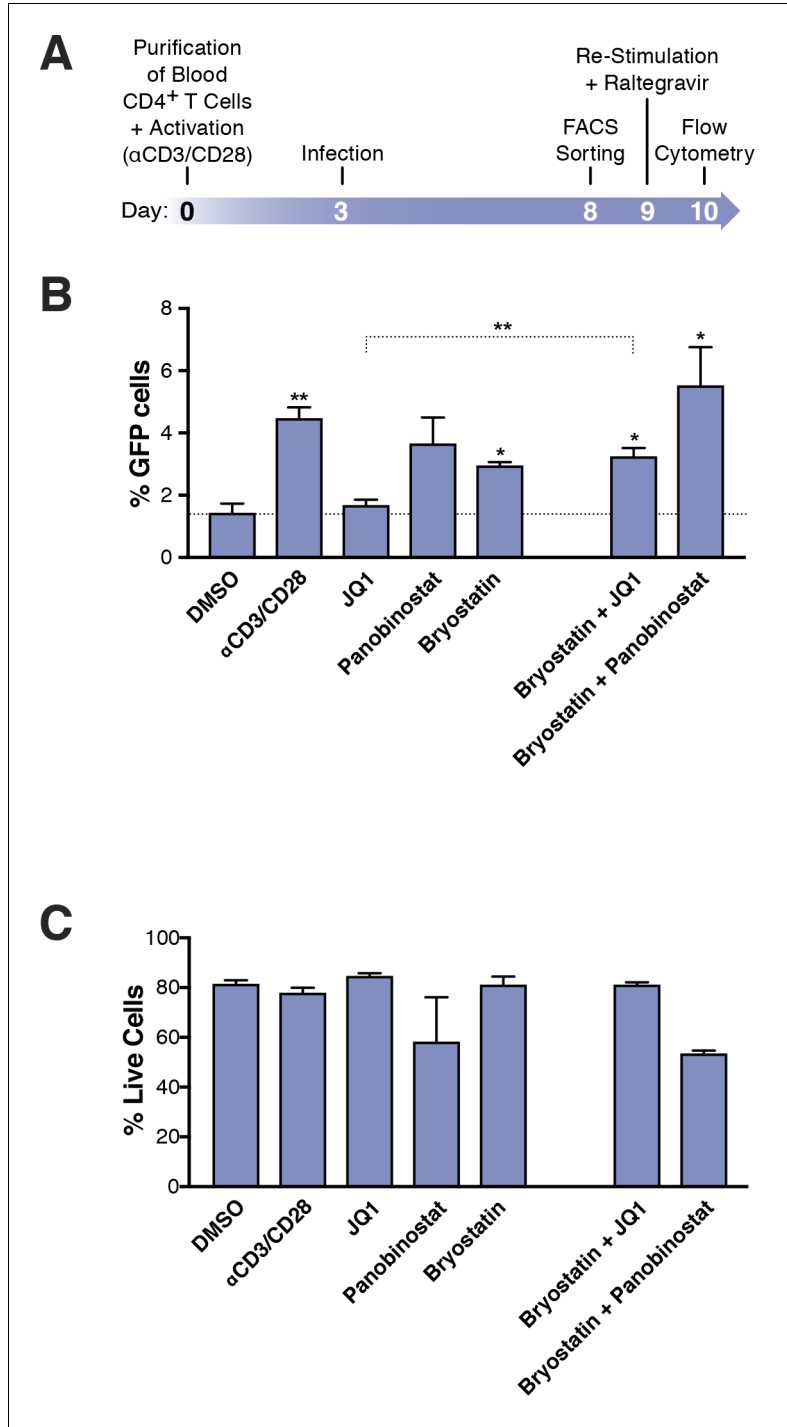

**Figure 3.** Few HIV_GKO latently infected primary CD4+ T cells are reactivated. (**A**) Schematic of experimental procedure with primary CD4+ T cells. Briefly, CD4+ T cells were purified from blood of healthy donors and activated for 72 hr with αCD3/CD28 beads and 100 U/ml IL-2 before infection with HIV_GKO. Five days post-infection, latently infected cells (csGFP- mKO2+) cells were sorted, put back in culture overnight and stimulated with different LRAs in presence of raltegravir for 24 hr before performing FACS analysis. (**B**) Percentage of GFP + cells is shown after stimulation of latently infected CD4+ T-cells with LRAs (n = 4 (different donors), mean +SEM, paired t-test) (**Figure 3—source data 1**). (**C**) Histogram plot of percent live cells for each drug treatment (n = 3 (different donors), mean + SEM, paired t-test) (**Figure 3—source data 1**). p-value: *p<0.05, **p<0.01 relative to DMSO.

DOI: https://doi.org/10.7554/eLife.34655.007

*Figure 3 continued on next page*

*Figure 3 continued*

The following source data is available for figure 3:

**Source data 1.** Percentage of GFP+ cells is shown after stimulation of latently infected CD4+ T-cells with LRAs as well as percent live cells for each drug treatment.
DOI: https://doi.org/10.7554/eLife.34655.008

overnight before restimulation with αCD3/CD28. After another 24 hr, cells were stained for the early, intermediate, and late markers of T cell activation CD69, CD25 and HLA-DR respectively. The three different populations, double negative, RLIC and NRLIC, had similar profiles of activated T-cell subsets, as shown in *Figure 4*, and were mainly composed of strongly activated cells (CD69+/CD25

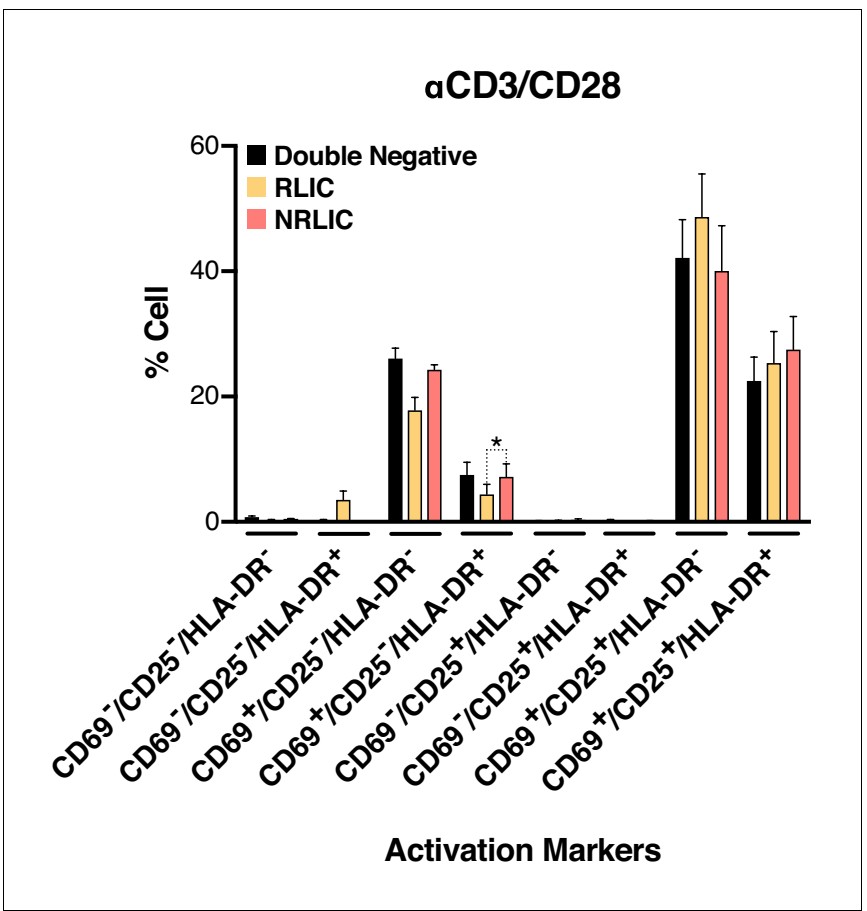

**Figure 4.** Low-level latency reactivation is not explained by low cellular responses to activation signals. T-cell activation patterns between double negative, reactivated (RLIC) and non-reactivated (NRLIC) latently infected cells. Briefly, CD4+ T-cells were purified from blood of four healthy donors and activated for 72 hr with αCD3/CD28 beads and 20 U/ml IL-2 before infection with HIV_{GKO}. At 4 days post-infection, csGFP- were sorted, cultured overnight and stimulated with αCD3/CD28 in presence of raltegravir. At 24 hr post-treatment, cells were stained for CD25, CD69, and HLA-DR activation markers before performing FACS analysis. (n = 4, mean +SEM, paired t-test; *p<0.05; **p<0.01) (*Figure 4—source data 1*).
DOI: https://doi.org/10.7554/eLife.34655.010

The following source data and figure supplement are available for figure 4:

**Source data 1.** CD25, CD69, and HLA-DR activation markers patterns between double negative, reactivated (RLIC) and non-reactivated (NRLIC) latently infected cells.
DOI: https://doi.org/10.7554/eLife.34655.012

**Figure supplement 1.** 24 hr treatment effectively activate primary CD4+ T cells.
DOI: https://doi.org/10.7554/eLife.34655.011

+/HLA-DR+/-). We only observed a statistically significant increase of NRLIC compared with RLIC in the CD69+/CD25-/HLA-DR+ population, however this small increase in a relatively minor population is insufficient to explain the low reactivation rate of latently infected cells. Overall, comparison of both reactivated and non reactivated latent populations showed little difference in their activation state.

## Integration sites, gene expression, transcription units and the fate of HIV infection

The role of the site of HIV integration into the genome in latency remains a subject of debate (*Chen et al., 2017*; *Dahabieh et al., 2014*; *Jordan et al., 2003*; *Jordan et al., 2001*; *Sherrill-Mix et al., 2013*). To identify possible differences in integration sites between reactivated and non-reactivated HIV genomes, primary CD4$^+$ T-cells were infected with HIV$_{GKO}$. At 5 days post-infection, productively infected cells (GFP+, PIC) were sorted and frozen. The GFP negative population (consisting of a mixture of latent and uninfected) was isolated and treated with αCD3/CD28. 48 hr post-induction, both non reactivated (NRLIC) and reactivated (RLIC) populations were isolated. Nine libraries (three donors, three samples/donor: PIC, RLIC, NRLIC) were constructed from genomic DNA as described (*Cohn et al., 2015*) and analyzed by high-throughput sequencing to locate HIV proviruses within the human genome. A total of 1803 virus integration sites were determined: 960 integrations in PIC, 681 in NRLIC, and 162 in RLIC (Integration Sites Source data).

To determine whether integration within genes differentially expressed during T-cell activation predicted infection reactivation fate, we compared our HIV integration dataset with a published dataset for gene expression in resting and activated (48 hr - αCD3/CD28) CD4$^+$ T cells from healthy individuals (*Ye et al., 2014*). The analysis revealed that most of the αCD3/CD28-induced latent proviruses were not integrated in genes responsive to T-cell activation signals (*Figure 5A and B*, *Figure 5—source data 1*). Interestingly, PIC and RLIC integration events were associated with genes whose basal expression was significantly higher than genes targeted in NRLIC, both in activated and resting T cells (*Figure 5C*, *Figure 5—source data 2*).

Next, we investigated whether different genomic regions were associated with productive, inducible or non-inducible latent HIV-1 infections. In agreement with previous studies (*Cohn et al., 2015*; *Dahabieh et al., 2014*; *Maldarelli et al., 2014*; *Wagner et al., 2014*), the majority of integration sites were found within genes in each population (*Figure 6A*, *Figure 6—source data 1*), although the proportion of genic integrations in NRLIC was significantly lower than in PIC and RLIC samples. Moreover, integration events in the PIC and RLIC populations were more frequent in transcribed regions (64% and 58%, respectively, [sum of low + medium + high transcribed regions] (*Figure 6B*), *Figure 6—source data 1*), while these regions were significantly less represented in the NRLIC (31%) (*Figure 6B*). As expected since introns represent a much larger proportion of genes, genic integration events were more frequent in the introns for each population (>65%, *Figure 6C*, *Figure 6—source data 1*). Finally, viral orientation of proviruses with respect to the transcriptional unit did not correlate with the fate of HIV infection (latent vs productive) or the reactivation or absence thereof of HIV latency (*Figure 6D*, *Figure 6—source data 1*).

## Chromatin modifications at the site of HIV integration and latency

Chromatin marks, such as histone post-translational modifications (e.g., methylation and acetylation) and DNA methylation, are involved in establishing and maintaining HIV-1 latency (*De Crignis and Mahmoudi, 2017*). We examined 500 bp regions centered on all integration sites in each population for several chromatin marks by comparing our data with several histone modifications and DNaseI ENCODE datasets. We first looked at distinct and predictive chromatin signatures, such as H3K4me1 (active enhancers), H3K36m3 (active transcribed regions), H3K9m3 and H3K27m3 (repressive marks of transcription) (reviewed in [*Kumar et al., 2015*; *Shlyueva et al., 2014*]). All three populations exhibited distinct profiles, although productive and inducible latent infections profiles appeared most similar (*Figure 7A*, *Figure 7—source data 1*). The analysis showed that PIC integration events were associated with active chromatin (i.e., transcribed genes - H3K36me3 or enhancers - H3K4me1), while NRLIC integration events appeared biased toward heterochromatin (H3K27me3 and H3K9me3) and non-accessible regions (DNase hyposensitivity).

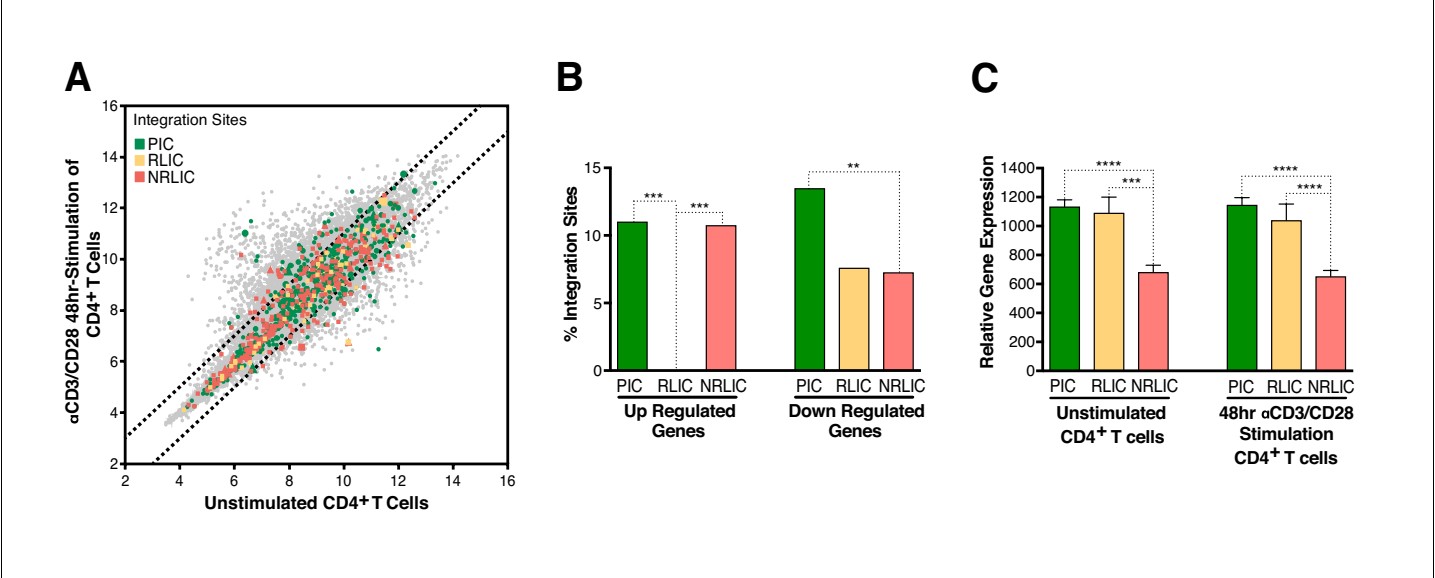

**Figure 5.** Relative expression of HIV-1 integration targeted genes for each population, before or after TCR activation. (A) Scatter chart showing primary CD4+ T-cell gene expression changes after 48 hr of stimulation with αCD3/CD28 beads. Integration sites displayed outside of the two solid gray lines were targeted genes whose expression is at least ± twofold differentially expressed after 48 hr stimulation. Plot points size can be different, the bigger the plot point is, the more integration events happened within the same gene. (B) Fraction of integration sites from the different populations PIC, RLIC or NRLIC, integrated within genes whose expression is at least ± twofold differentially expressed after 48 hr of αCD3/CD28 stimulation (**p<0.01; ***p<0.001; two-proportion z test) (*Figure 5—source data 1*). (C) Relative expression of genes targeted by HIV-1 integration in PIC, RLIC or NRLIC before TCR stimulation and after αCD3/CD28 stimulation (n = 3, mean +SEM, paired t-test). ***p<0.001; ****p<0.0001. (*Figure 5—source data 2*).
DOI: https://doi.org/10.7554/eLife.34655.013

The following source data is available for figure 5:

**Source data 1.** Fraction of integration sites from the different populations PIC, RLIC or NRLIC, integrated within genes whose expression is at least ± twofold differentially expressed after 48 hr of αCD3/CD28 stimulation.
DOI: https://doi.org/10.7554/eLife.34655.014

**Source data 2.** Relative expression of genes targeted by HIV-1 integration in PIC, RLIC or NRLIC before TCR stimulation and after 48 hr αCD3/CD28 stimulation.
DOI: https://doi.org/10.7554/eLife.34655.015

Marini *et al.* recently reported that HIV-1 mainly integrates at the nuclear periphery (*Marini et al., 2015*). We therefore examined the topological distribution of integration sites from each population inside the nucleus by comparing our integration site data with a previously published dataset of lamin-associated domains (LADs) (*Guelen et al., 2008*). LADs consist of H3K9me2 heterochromatin and are present at the nuclear periphery. This analysis showed that latent integration sites from both RLIC and NRLIC were in LADs to a significantly higher degree (32% and 30.4%) than productive integrations (23.6%) (p<0.05, *Figure 7B*, *Figure 7—source data 1*). Overall, these data show similar features between productively infected cells and inducible latently infected cells, while non-reactivated latently infected cells appear distinct from the other populations. These findings support a prominent role for the site of integration and the chromatin context for the fate of the infection itself, as well as for latency reversal.

## Discussion

Dual-color HIV-1 reporters are unique and powerful tools (*Calvanese et al., 2013*; *Dahabieh et al., 2013*), that allow for the identification and the isolation of latently infected cells from productively infected cells and uninfected cells. Latency is established very early in the course of HIV-1 infection (*Archin et al., 2012b*; *Chun et al., 1998*; *Whitney et al., 2014*) and, until the advent of dual-reporter constructs, no primary HIV-1 latency models have allowed the study of latency heterogeneity at this very early stage. Importantly, the comparison of data obtained from distinct primary HIV-1

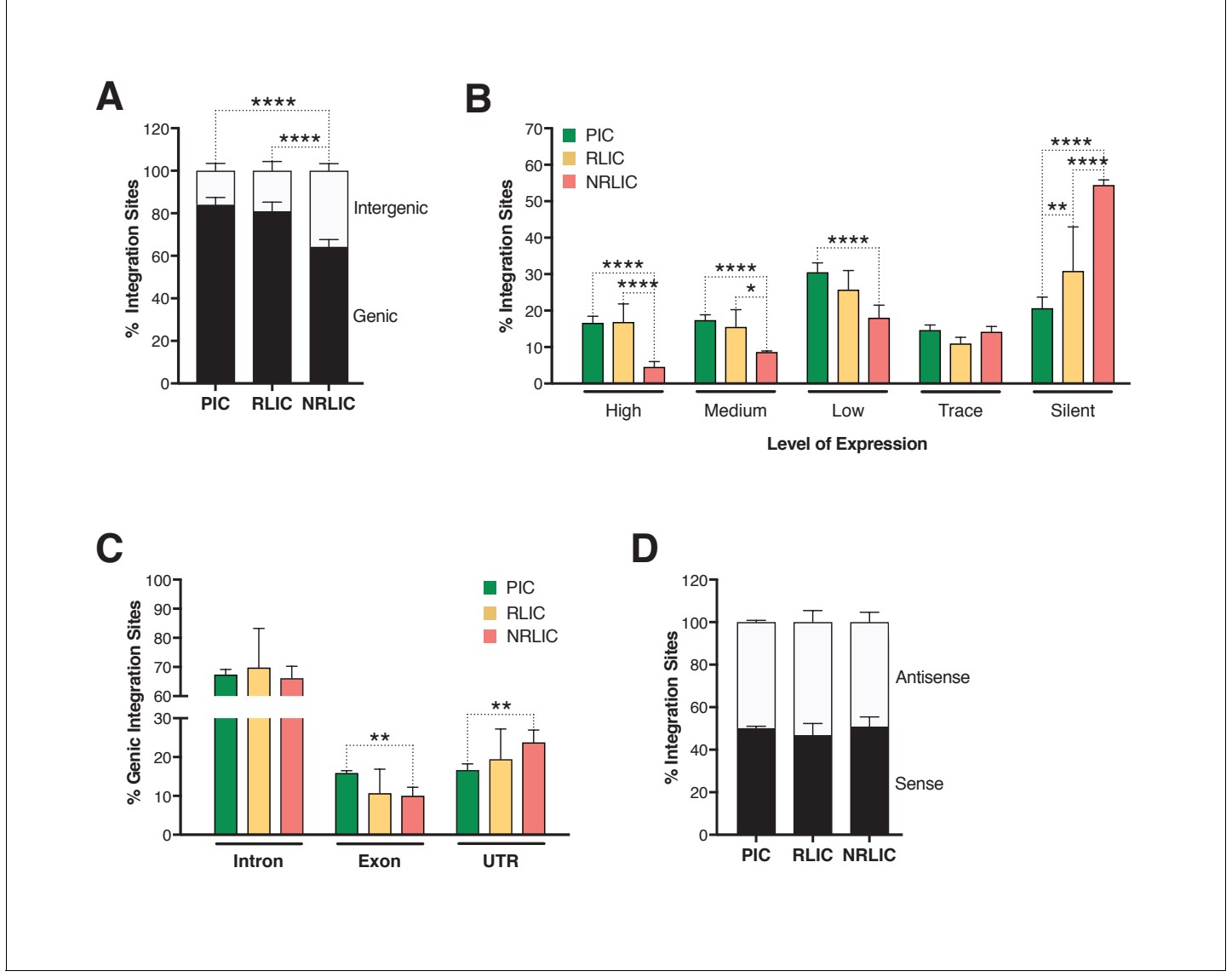

**Figure 6.** Insertion landscapes of HIV-1. (**A**) Proportion of mapped insertions that are in genic or intergenic regions. (*Figure 6—source data 1*). (**B**) Proportion of integration sites in transcribed regions with high (top 1/8), medium (top 1/4–1/8), low expression (top 1/2–1/4), trace (bottom 1/2) or silent (0) expression. (*Figure 6—source data 1*). (**C**) Proportion of unique genic integration sites located in introns, exons, UTR or promoters. (*Figure 6—source data 1*). (**D**) Transcriptional orientation of integrated HIV-1 relative to host gene. (*Figure 6—source data 1*). p-value: *p<0.05; **p<0.01; ***p<0.001; ****p<0.0001 using two-proportion z test.

DOI: https://doi.org/10.7554/eLife.34655.016

The following source data is available for figure 6:

**Source data 1.** Proportion of mapped insertions that are in genic or intergenic regions; of integration sites in transcribed regions with high, medium, low expression, trace or silent expression; of unique genic integration sites located in introns, exons, UTR or promoters; and transcriptional orientation of integrated HIV-1 relative to host gene.

DOI: https://doi.org/10.7554/eLife.34655.017

latency models is complicated as some models are better suited to detect latency establishment (e.g., dual-reporters), while others are biased towards latency maintenance (e.g., Bcl2-transduced CD4+ T cells). The use of env-defective viruses limits HIV replication to a single-round and, thereby limits the appearance of defective viruses (*Bruner et al., 2016*).

In this study, we describe and validate an improved version of HIV$_{DuoFluoI}$, previously developed in our laboratory (*Calvanese et al., 2013*), which accurately allows for: (a) the quantification of

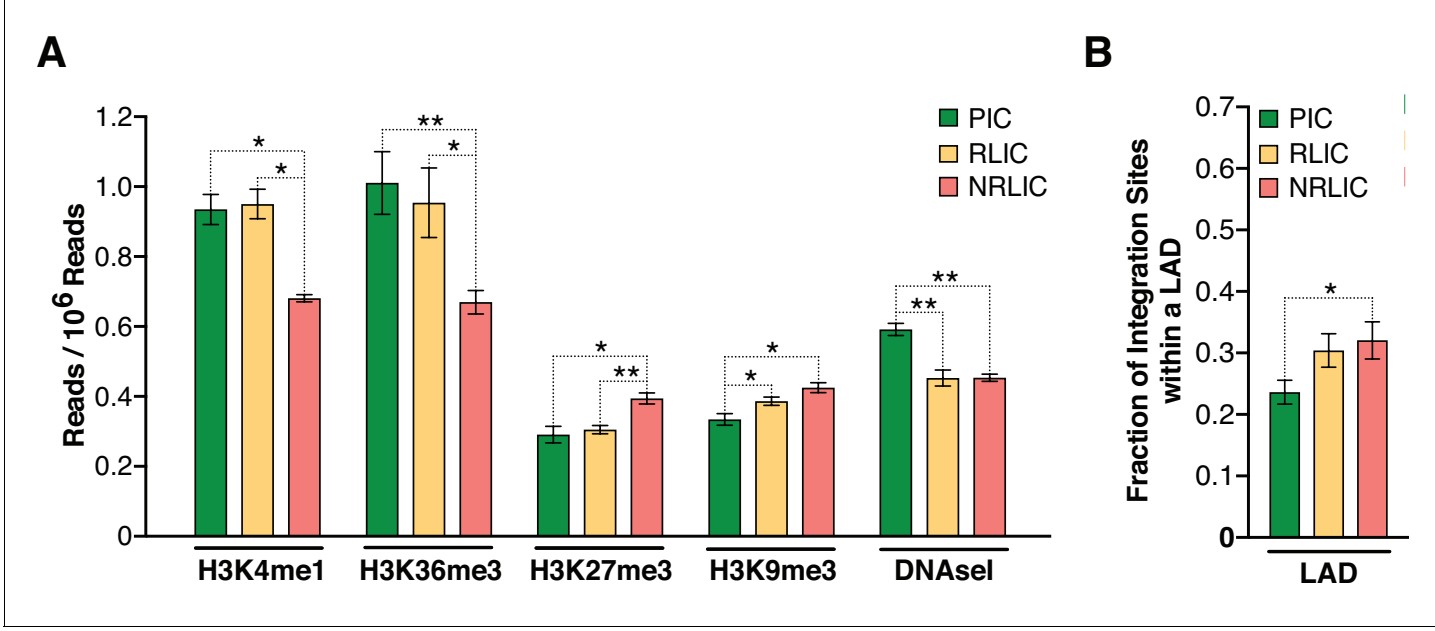

**Figure 7.** Epigenetics marks and nuclear localization of HIV-1 integration sites. (**A**) 500 bp centered on HIV-1 integration sites for each population were analyzed for the presence of H3K4me1 (active enhancers), H3K36m3 (active transcribed regions), H3K9m3 and H3K27m3 (repressive marks of transcription), and DNA accessibility (DNAseI). (*Figure 7—source data 1*). (**B**) Nuclear localization of HIV-1 integration sites. Quantification was based on inside a LAD (=1) or outside (=0), which means the Y axis represents the fraction of integrations within a LAD. (*Figure 7—source data 1*). (n = 3–4 ENCODE donors, mean +SEM, paired t-test). *p<0.05; **p<0.01; ***p<0.001).

DOI: https://doi.org/10.7554/eLife.34655.018

The following source data is available for figure 7:

**Source data 1.** HIV-1 integration sites for each population were analyzed for the presence of H3K4me1, H3K36m3, H3K9m3, H3K27m3, DNA accessibility, as well as their nuclear localization.

DOI: https://doi.org/10.7554/eLife.34655.019

latently infected cells, (b) the purification of latently infected cells, and (c) the evaluation of the 'shock and kill' strategy. Our data highlight two important facts: (a) cell-associated HIV RNA quantification does not reflect the number of cells undergoing viral reactivation, and (b) a small portion of the cells carrying latent proviruses (<5%) is reactivated, although LRAs target the whole latent population. Hence, even if cells harboring reactivated virus die, this small reduction would likely remain undetectable when quantifying the latent reservoir in vivo. Our data are in agreement with previous reports, which show that levels of cellular HIV RNA and virion production are not correlated, and that the absolute number of cells being reactivated by αCD3/CD28 is indeed limited to a small fraction of latently infected cells (*Cillo et al., 2014*; *Sanyal et al., 2017*; *Yucha et al., 2017*). Using our dual-fluorescence reporter, we confirm these findings, and extend these observations to LRAs combinations. However, although LRAs combinations show synergy when measuring cell-associated HIV RNA, we do not find such synergy at the level of individual cells, but rather only partial additive effect. Our work, as well as that of others (*Cillo et al., 2014*; *Sanyal et al., 2017*; *Yucha et al., 2017*), demonstrate the importance of single cell analysis when it comes to the evaluation of potential LRAs. Indeed, it is necessary to determine wheter potential increases in HIV RNA after stimulation in a bulk population result from a small number of highly productive cells, or from a larger but less productive population, as these two mechanisms likely have very different impacts on the latent reservoir.

Our data further highlight the heterogeneous nature of the latent reservoir (*Chen et al., 2017*; *Ho et al., 2013*). We currently have a limited understanding of why some latently infected cells are capable of being induced while others are not. It is possible that different chromatin environments impose different degrees of transcriptional repression on the integrated HIV genome, with the non reactivatable latent HIV corresponding to the most repressive environment. (*Chen et al., 2017*).

Since HIV$_{GKO}$ allows for the isolation of productively infected cells and reactivated latent cells from those that do not reactivate, it provides a unique opportunity to explore the impact of HIV integration on the fate of the infection.

Different integration site-specific features contribute to latency, such as the chromatin structure, including adjacent loci but also the provirus location in the nucleus (*Lusic and Giacca, 2015*; *Lusic et al., 2013*). Viral integration is a semi-random process (*Bushman et al., 2005*) in which HIV-1 preferentially integrates into active genes (*Barr et al., 2006*; *Bushman et al., 2005*; *Demeulemeester et al., 2015*; *Ferris et al., 2010*; *Han et al., 2004*; *Lewinski et al., 2006*; *Mitchell et al., 2004*; *Schröder et al., 2002*; *Sowd et al., 2016*; *Wang et al., 2007*). LEDGF, one of the main chromatin-tethering factors of HIV-1, binds to the viral integrase and to H3K36me3, and to a lesser extent to H3K4me1, thus directing the integration of HIV-1 into transcriptional units (*Daugaard et al., 2012*; *Eidahl et al., 2013*; *Pradeepa et al., 2012*). Also CPSF6, which binds to the viral capsid, markedly influences integration into transcriptionally active genes and regions of euchromatin (*Sowd et al., 2016*), explaining how HIV-1 maintains its integration in the euchromatin regions of the genome independently of LEDGF (*Quercioli et al., 2016*). Several studies have characterized the integration sites, however, these analyses have been restricted to productive infections.

Using ENCODE reference datasets, our data are consistent with previous results, showing that HIV-1 preferentially targets actively transcribed regions (*Marini et al., 2015*; *Wang et al., 2007*; *Chen et al., 2017*). However, non-inducible latent proviruses are observed to be integrated to a higher extent into silenced chromatin. In addition, even though HIV integration is normally strongly disfavored in the heterochromatic condensed regions in LADs due to low chromatin accessibility, we show that some HIV integration does occur in LADs when using a previously published dataset of LADs (*Guelen et al., 2008*; *Marini et al., 2015*), and that latent proviruses that are not readily reactivatable are integrated at higher extent in LADs.

Importantly, we identify a unique rare population among the latent cells that can be reactivated. In contrast to the non-inducible latent infections, the latency reversal of inducible latent proviruses might be explained by integration in an open chromatin context, similar to integration sites for productive proviruses, followed by subsequent heterochromatin formation and proviral silencing. As a consequence, the distinct integration sites between induced and non-induced latent proviruses highlight new possibilities for cure strategies. Indeed, the 'shock and kill' strategy aims to reactivate and eliminate every single replication-competent latent provirus, since a single remaining cell carrying a latent inducible provirus could, in theory, reseed the infection. However, our study and others point out several significant barriers to successful implementation of the 'shock and kill' strategy. First, LRAs only reactivate a limited fraction of latent proviruses. It is likely that some of the non-induced proviruses, such as those integrated into enhancers and transcriptionnal active regions of the genome, will reactivate after several rounds of activation, due to the stochastic nature of HIV activation (*Dar et al., 2012*; *Ho et al., 2013*; *Singh et al., 2010*; *Weinberger et al., 2005*). It is also likely that better suited LRAs combinations (two or more LRAs) will reactivate some of the non-induced proviruses integrated into silenced chromatin marked by H3K27me3 and H3K9me3. Indeed, several studies have shown that the pharmaceutical inhibition of H3K27me3 and H3K9me2/3 could sensitize latent proviruses to LRAs (*Friedman et al., 2011*; *Nguyen et al., 2017*; *Tripathy et al., 2015*). Second, Shan *et al*. have shown that latently reactivated cells are not cleared due to cytopathic effects or CTL response implying that immunomodulatory approaches, in addition of more potent LRAs, are likely required to achieve a cure for HIV infection (*Shan et al., 2012*).

In conclusion, the heterogeneity of the latent reservoir calls for therapies addressing the different pools of latently infected cells. While 'shock and kill' might be helpful in reactivating and possibly eliminating a small subset of highly reactivatable latent HIV genomes, other approaches will be necessary to control or eliminate the less readily reactivatable population identified here and in patients. Perhaps, this latter population should rather be 'blocked and locked' using latency-promoting agents (LPAs), as described by several groups (*Besnard et al., 2016*; *Kessing et al., 2017*; *Kim et al., 2016*; *Vranckx et al., 2016*). For a functional cure, a stably silenced, non-reactivatable provirus is preferable to a lifetime of chronic active infection.

## Materials and methods

### Patients' samples

Four HIV-1-infected individuals, who met the criteria of suppressive ART, undetectable plasma HIV-1 RNA levels (<50 copies/ml) for a minimum of six months, and with CD4$^+$ T cell count of at least 350 cells/mm$^3$, were enrolled. The participants were recruited from the SCOPE cohort at the University of California, San Francisco. *Table 1* details the characteristics of the study participants.

### Plasmids construction

To construct HIV$_{GKO}$, the csGFP sequence was designed and ordered from Life Technologies. The sequence was cut out from Life Technologies' plasmid with BamHI and XhoI and cloned into Duo-FluoI, previously cut with the same enzymes (DuoFluoI-csGFP). HIV$_{GKO}$ was creating by PCR overlapping: csGFP-EF1α (Product 1) was PCR amplified from DuoFluoI using primers P1: 5' for-GATTAG TGAACGGATCCTTGGCAC-3' and P2: 5' rev-GGCTTGATCACAGAAACCATGGTGGCGACCGG TAGCGC-3'. mKO2 (Product 2) was PCR amplified from Brian Webster's plasmid (kind gift from Warner Greene) using primers P3: 5' for-GCGCTACCGGTCGCCACCATGGTTTCTGTGATCAA GCC-3' and P4: 5' rev-CTCCATGTTTTTCCAGGTCTCGAGCCTAGCTGTAGTGGGC CACGGC-3'. Finally, we amplified the 3'LTR sequence (Product 3) from RGH plasmid (*Dahabieh et al., 2013*) using primers P5: 5' for-GCTCGAGACCTGGAAAAACATGGAG-3' and P6: 5' rev-GTGCCACC TGACGTCTAAGAAACC-3', to add a fragment containing the AatII restriction site, in order to ligate the csGFP-EF1α-mKO2 cassette into pLAI (*Peden et al., 1991*). We then did sequential PCRs: products 1 and 2 were amplified using primers P1 and P4. PCR product (1 + 2) was mixed with product three and PCR amplified with P1 and P6 thus creating the full cassette. The cassette csGFP-EF1α-mKO2 was then digested with BamHI and AatII, and cloned into pLAI previously digested with the same enzymes to create HIV$_{GKO}$.

Of note, the Envelope open reading frame was disrupted by the introduction of a frame shift at position 7136 by digestion with KpnI, blunting, and re-ligation.

To construct GKO-ΔU3 3'LTR, we cloned a ΔU3 linker from pTY-EFeGFP (*Chang et al., 1999*; *Cui et al., 1999*; *Iwakuma et al., 1999*; *Zolotukhin et al., 1996*) into the KpnI/SacI sites of the 3' LTR in HIV$_{GKO}$.

### Virus production

The production of HIV$_{GKO}$ and the assessment of HIV Latency Reversal Agents in Human Primary CD4+ T Cells are described in more detail at Bio-protocol (*Battivelli and Verdin, 2018*). Pseudo-typed HIV$_{DuoFluoI}$ and HIV$_{GKO}$ viral stocks were generated by co-transfecting (standard calcium phosphate transfection method) HEK293T cells with a plasmid encoding HIV$_{DuoFluoI}$ or HIV$_{GKO}$, and a plasmid encoding HIV-1 dual-tropic envelope (pSVIII-92HT593.1). Medium was changed 6–8 hr post-transfection, and supernatants were collected after 48 hr, centrifuged (20 min, 2000 rpm, RT), filtered through a 0.45 µM membrane to clear cell debris, and then concentrated by ultracentrifugation (22,000 g, 2 hr, 4°C). Concentrated virions were resuspended in complete media and stored at −80°C. Virus concentration was estimated by p24 titration using the FLAQ assay (*Gesner et al., 2014*).

### Primary cell isolation and cell culture

CD4$^+$ T cells were extracted from peripheral blood mononuclear cells (PBMCs) from continuous-flow centrifugation leukophoresis product using density centrifugation on a Ficoll-Paque gradient (GE Healthcare Life Sciences, Chicago, IL). Resting CD4$^+$ lymphocytes were enriched by negative depletion with an EasySepHuman CD4$^+$ T Cell Isolation Kit (Stemcell Technologies, Canada). Cells were cultured in RPMI medium supplemented with 10% fetal bovine serum, penicillin/streptomycin and 5 µM saquinavir.

Primary CD4$^+$ T cells were purified from healthy donor blood (Blood Centers of the Pacific, San Francisco, CA, and Stanford Blood Center), by negative selection using the RosetteSep Human CD4$^+$ T Cell Enrichment Cocktail (StemCell Technologies, Canada). Purified resting CD4$^+$ T cells from HIV-1 or healthy individuals were cultured in RPMI 1640 medium supplemented with 10% FBS, L-glutamine (2 mM), penicillin (50 U/ml), streptomycin (50 mg/ml), and IL-2 (20 to 100 U/ml) (37°C,

5% $CO_2$). Spin-infected primary CD4$^+$ T cells were maintained in 50% of complete RPMI media supplemented with IL-2 (20–100 U/ml) and 50% of supernatant from H80 cultures (previously filtered to remove cells) without beads. Medium was replenished every 2 days until further experiment.

HEK293T cells were obtained from ATCC (mycoplasma free). Feeder cells H80 was a kind gift from Jonathan Karn. H80 cells were cultured in RPMI 1640 medium supplemented with 10% fetal bovine serum (FBS), L-glutamine (2 mM), penicillin (50 U/ml), and streptomycin (50 mg/ml) (37°C, 5% $CO_2$). HEK293T cells were cultured in DMEM medium supplemented with 10% FBS, 50 U/ml penicillin, and 50 mg/ml streptomycin.

## Cell infection

Purified CD4$^+$ T cells isolated from healthy peripheral blood were stimulated with αCD3/CD28 activating beads (Thermofisher, Waltham, MA) at a concentration of 0.5 bead/cell in the presence of 20–100 U/ml IL-2 (PeproTech, Rocky Hill, NJ) for three days. All cells were spinoculated with either HIV$_{DuoFluoI}$, HIV$_{GKO}$ or HIV Δ3U-GKO at a concentration of 300 ng of p24 per $1.10^6$ cells for 2 hr at 2000 rpm at 32°C without activation beads.

Infected cells were either analyzed by flow cytometry or sorted 4–5 days post-infection.

## Latency-reversing agent treatment conditions

CD4$^+$ T cells were stimulated for 24 hr unless stipulated differently, with latency-reversing agents at the following concentrations for all single and combination treatments: 10 nM bryostatin-1, 1 μM JQ1, 30 nM panobinostat, αCD3/CD28 activating beads (1 bead/cell), or media alone plus 0.1% (v/v) DMSO. For all single and combination treatments, 30 μM Raltregravir (National AIDS Reagent Program) was added to media. Concentrations were chosen based on Laird et al. paper (*Laird et al., 2015*).

## Staining, flow cytometry and cell sorting

Cells from *Figure 4* were stained with α-CD69-PE-Cy7 (561928), α-CD25-APC (560987), and α-HLA-DR-PerCP-Cy5.5 (562007) (BD Bioscience, Franklin Lakes, NJ).

Before collecting data using the FACS LSRII (BD Biosciences, Franklin Lakes, NJ) or the FACS AriaII (BD Biosciences, Franklin Lakes, NJ, *Figures 3* and *4*), cells were stained with violet Live/Dead Fixable Dead Cell Stain (Thermofisher, Waltham, MA) and fixed with 2% formaldehyde. Analyses were performed with FlowJo V10.1 software (TreeStar).

Sorting of infected CD4$^+$ T cells was performed with a FACS AriaII (BD Biosciences, Franklin Lakes, NJ) based on their GFP and mKO2 fluorescence markers at 4/5 days post-infection, and placed back in culture for further experimentation. In the experiments shown in *Figures 2B* and *4*, we isolated both HIV$_{GKO}$ latently infected cells (GFP-, mKO2+, 3%) and uninfected cells (csGFP-, mKO2-, 97%) five days post-infection, before treating cells with LRAs.

In the experiment shown in *Figure 3*, we isolated pure latent cells (GFP-, mKO2+) five days post-infection, before treating this pure population with LRAs.

## DNA, RNA and protein extraction, qPCR and western blot

RNA and proteins (*Figure 1B and C*) were extracted with PARIS$^{TM}$ kit (Ambion, Thermofisher, Waltham, MA) according to manufacturer's protocol from same samples. RNA was retro-transcribed using random primers with the SuperScript II Reverse Transcriptase (Thermofisher, Waltham, MA) and qPCR was performed in the AB7900HT Fast Real-Time PCR System, using 2X HoTaq Real Time PCR kit (McLab, South San Francisco, CA) and the appropriate primer-probe combinations described in (*Calvanese et al., 2013*). Quantification for each qPCR reaction was assessed by the ddCt algorithm, relative to Taq Man assay GAPDH Hs99999905_m1. Protein content was determined using the Bradford assay (Bio-Rad, Hercules, CA) and 20 μg were separated by electrophoresis into 12% SDS-PAGE gels. Bands were detected by chemiluminescence (ECL Hyperfilm Amersham, GE Healthcare Life Sciences, Chicago, I) with anti-Vif, HIV-p24 and α-actin (Sigma, Saint-Louis, MO) primary antibodies.

Total RNA (*Figure 2A and B*) wasextracted using the Allprep DNA/RNA/miRNA Universal Kit (Qiagen, Germany) with on-column DNAase treatment (Qiagen RNase-Free DNase Set, Germany). cDNA synthesis was performed using SuperScript IV Reverse Transcriptase with a combination of

random hexamers and oligo-dT primers (ThermoFisher, Waltham, MA). Relative cellular HIV mRNA levels were quantified using a qPCR TaqMan assay using primers and probes described in (*Bullen et al., 2014*) on a QuantStudio 6 Flex Real-Time PCR System (Thermofisher, Waltham, MA). Relative cell-associated HIV mRNA copy numbers were determined in a reaction volume of 20 μL with 10 μL of 2x TaqMan Universal Master Mix II with UNG ( Thermofisher, Waltham, MA), 4 pmol of each primer, 4 pmol of probe, 0.5 μL reverse transcriptase, and 2.5 μL of cDNA. Cycling conditions were 50℃ or 2 min, 95℃ for 10 min, then 60 cycles of 95℃ for 15 s and 60℃ for 1 min. Real-time PCR was performed in triplicate reaction wells, and relative cell-associated HIV mRNA copy number was normalized to cell equivalents using human genomic GAPDH expression by qPCR and applying the comparative Ct method (*Livak and Schmittgen, 2001*).

## HIV integration site libraries and computational analysis

HIV integration site libraries and computational analysis were executed in collaboration with Lilian B. Cohn and Israel Tojal Da Silva as described in their published paper (*Cohn et al., 2015*), with a few small changes added to the computational analysis pipeline. First, we included integration sites with only a precise junction to the host genome. Second, to eliminate any possibility of PCR mispriming, we have excluded integration sites identified within 100 bp (50 bp upstream and 50 bp downstream) of a 9 bp motif identified in our LTR1 primer: TGCCTTGAG. Thirdly we have merged integration sites within 250 bp and have counted each integration site as a unique event. The list of integration sites for each donor and each population can be found as a source data file linked to this manuscript (Integration Sites *Source data 1*).

## Datasets

Chromatin data (ChIP-seq) from CD4$^+$ T cells was downloaded from ENCODE: H3K4me1 (ENCFF112QDR, ENCFF499NFE, ENCFF989BNS), H3K9me3 (ENCFF044NLN, ENCFF736KRZ, ENCFF844IWD, ENCFF929BPC), H3K27ac (ENCFF618IUD, ENCFF862SKP), H3K27me3 (ENCFF124QDD, ENCFF298JKA, ENCFF717ODY), H3K36me3 (ENCFF006VTQ, ENCFF169QYM, ENCFF284PKI, ENCFF504OUW), DNAse (GSM665812, GSM665839, GSM701489, GSM701491). Data were analyzed using Seqmonk (v0.33, http://www.bioinformatics.bbsrc.ac.uk/projects/ seq-monk/).

We calculated expression (GSM669617) and chromatin mark abundance (the remaining ENCODE datasets) at the integration sites as bins of 500 bp centered on the integration site (read count quantification in Seqmonk: all non-duplicated reads regardless of strand, corrected per million reads total, non-log transformed). Gene annotations were not taken into account. Thresholds for expression values (upper 1/8th, upper quarter, half, and above 0) were set to distinguish five different categories, set as the upper 1/8th of expression values (high), upper quarter–1/8th (medium), upper half–quarter (low), lower half but above 0 (trace), 0 (silent).

CD4$^+$ T cells activation data in *Figure 5A* weredownloaded from GEO (GSE60235).

## Statistical analysis

Significance was analyzed by either paired t-test (GraphPad Prism) or proportion test (standard test for the difference between proportions), also known as a two-proportion z test (https://www.medcalc.org/calc/comparison_of_proportions.php), and specified in the manuscript.

## Acknowledgements

We thank Herb Kasler for critical reading of the manuscript and help with data analysis, Giovanni Maki, Teresa Roberts and John Carroll for graphic preparation, Gary Howard for editorial assistance, and Veronica Fonseca for administrative assistance. EB was supported by a post-doctoral fellowship from UCSF CFAR and a CHRP fellowship. MD was supported by a CIHR 201311MFE-321128–179658. EV was supported by funds from NIH 1R01DA030216, 1DP1DA031126, NIH/NIAID R01Ai117864 NIH/NIDA/1R01DA041742-01, NIH/NIDCR/1R01DE026010-01, and 5–31532. We would also like to thank Marielle Cavrois and Herb Kasler, respectively directors of the Gladstone and Buck Flow Cores. The Gladstone Flow Core was funded by NIH Grants P30AI027763 and S10 RR028962 and by the University of California, San Francisco-Gladstone Institute of Virology and Immunology Center for AIDS Research (CFAR). The standards for PCR methods were made available

with help from the University of California San Francisco-Gladstone Institute of Virology and Immunology Center for AIDS Research (CFAR), an NIH-funded program (P30 AI027763), and NIH/NIAIDR21AI129636 for MAM. We thank the amfAR Institute for HIV Cure Research. JPS. was supported by the Swedish Research Council (VR2015-02312) and Cancerfonden (CAN2016/576). SKP was supported by a NIGMS fund R01GM117901

## Additional information

### Funding

| Funder | Grant reference number | Author |
|---|---|---|
| Center for AIDS Research, University of California, San Diego | | Emilie Battivelli Mohamed Abdel-Mohsen |
| California HIV/AIDS Research Program | | Emilie Battivelli |
| Canadian Institutes of Health Research | 201311MFE-321128-179658 | Matthew S Dahabieh |
| National Institute of Allergy and Infectious Diseases | R21AI129636 | Mohamed Abdel-Mohsen |
| Svenska Forskningsrådet Formas | VR2015-02312 | J Peter Svensson |
| Cancerfonden | CAN2016/576 | J Peter Svensson |
| National Institute of General Medical Sciences | R01GM117901 | Satish K Pillai |
| National Institute of Allergy and Infectious Diseases | R01Ai117864 | Eric Verdin |
| National Institute on Drug Abuse | 1R01DA041742-01 | Eric Verdin |
| National Institute of Dental and Craniofacial Research | 1R01DE026010-01 | Eric Verdin |
| National Institute of Dental and Craniofacial Research | 5-31532 | Eric Verdin |

The funders had no role in study design, data collection and interpretation, or the decision to submit the work for publication.

### Author contributions

Emilie Battivelli, Conceptualization, Resources, Data curation, Formal analysis, Supervision, Funding acquisition, Validation, Investigation, Visualization, Methodology, Writing—original draft, Project administration, Writing—review and editing; Matthew S Dahabieh, Methodology, Writing—review and editing; Mohamed Abdel-Mohsen, Resources, Investigation, Writing—review and editing; J Peter Svensson, Conceptualization, Resources, Data curation, Formal analysis, Validation, Writing—original draft, Writing—review and editing; Israel Tojal Da Silva, Data curation, Software, Formal analysis, Methodology; Lillian B Cohn, Resources, Methodology, Writing—review and editing; Andrea Gramatica, Warner C Greene, Satish K Pillai, Resources, Writing—review and editing; Steven Deeks, Resources; Eric Verdin, Conceptualization, Supervision, Funding acquisition, Validation, Methodology, Writing—review and editing

### Author ORCIDs

Emilie Battivelli  http://orcid.org/0000-0003-3297-0231
J Peter Svensson  http://orcid.org/0000-0002-5863-6250
Lillian B Cohn  http://orcid.org/0000-0002-3485-8692
Steven Deeks  http://orcid.org/0000-0001-6371-747X
Eric Verdin  http://orcid.org/0000-0003-3703-3183

**Decision letter and Author response**
Decision letter https://doi.org/10.7554/eLife.34655.027
Author response https://doi.org/10.7554/eLife.34655.028

## Additional files

### Supplementary files
• Source data 1. Integration Sites - Source Data: List of integration sites for each donor and each population.
DOI: https://doi.org/10.7554/eLife.34655.020

• Transparent reporting form
DOI: https://doi.org/10.7554/eLife.34655.021

### Data availability
All sequencing data generated during this study are included in the Integration sites Source data file 1

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
