## [Decision Letter]

Thank you for submitting your article "Chromatin Functional States Correlate with HIV Latency Reversal in Infected Primary CD4^+^ T Cells" for consideration by *eLife*. Your article has been reviewed by three peer reviewers, and the evaluation has been overseen by a Reviewing Editor and Prabhat Jha as the Senior Editor. The following individuals involved in review of your submission have agreed to reveal their identity: Marina Lusic (Reviewer #1); Vicente Planelles (Reviewer #2).

The reviewers have discussed the reviews with one another and the Reviewing Editor has drafted this decision to help you prepare a revised submission.

Summary:

In this elegant study, the authors report on the establishment of latency and reactivation of latent proviruses using a dual reporter HIV-1 virus (HIV_GKO_), which allows accurate separation and quantification of cells actively producing HIV from those harboring LTR silent, latent viruses.

By using several latency-reversing agents (LRAs) the authors compare the efficacy of HIV-1 reactivation in cells from HIV-1 infected individuals and in cells infected in vitro with the HIV_GKO_. HDAC inhibitor panobinostat, BRD4 inhibitor JQ1, and the PKC inhibitor bryostatin-1 showed limited reactivation unless used in combination. The similar trends in reactivation potential with single or combination LRAs recapitulates well the existing literature. The observation that the synergistic effect of these drugs in reactivating HIV in patient samples was stronger than that observed in the cells infected in vitro is very interesting.

The authors mapped a total of 1,803 integration sites mostly located in introns of actively transcribing genes. The transcription profiles of these genes as well as several histone marks using ENCODE data were also analyzed. Proviruses appear to be integrated in intragenic regions, with proviruses integrated more frequently in enhancer and actively transcribed regions of the genome compared to silenced regions. Non-inducible latently infected cells harbored virus integrated in enhancer and activated regions to a higher extent than in silenced regions, although reduced compared to productively infecting and reactivated proviruses. Non-inducible latently infected cells showed an increased level of integration into heterochromatin compared to productively and reactivated proviruses. The authors conclude that HIV-1 reversal from latency highly depends on the host transcriptional activity and chromatin context. Integration sites from PICs are outside of lamin-associated domains (LADs) whereas those from RLIC and NRLIC are found more often within LADs compared to latent cells.

In conclusion, this well-written manuscript highlights the heterogeneous nature of the HIV-1 latent reservoir and the challenges it poses to strategies seeking to induce virus from latency as a method to eradicate HIV infection. Moreover, it underlines several important aspects regarding latent HIV, such as that viral RNA does not truly reflect the number of cells that are reactivated and that LRAs and T cell stimulation with CD3/CD28 results in reactivation of only a small portion of latent viruses. This powerful new dual color HIV reagent will be exciting to those in the field.

A revised manuscript will be considered if the authors can address the major points outlined below within a reasonable amount of time.

Essential revisions:

1) Throughout the paper the definition of productively infected cells appears to change. For example, in Figure 1D the ratio of latent to productively infected cells, appears to be calculated by combining csGFP+mKO2- and csGFP+mKO2+ populations whereas in the text corresponding to Figure 1E and 1F the productively infected population is defined as csGFP+mKO2+ cells only. While the csGFP+mKO2- population is reduced in the HIV_GKO_ model, this population is not negligible and should be consistently included when examining the productively infected population.

2) As the authors aim to establish a superior HIV latency model they should directly quantify the percent increase in latently infected cells compared to their previous DuoFluo reporter instead of only stating that they can identify a 'much larger' number of latent cells.

3) Throughout the paper the authors express surprise at the low level of reactivation of the latent reservoir observed while also stating that this observation is in agreement with the current literature giving a confusing impression of the novelty of their results. Please amend.

4) The fact that JQ1 and bryostatin induce reactivation in patient samples but only little effect on in vitro infected cells is a very interesting finding, pointing to differences in the sites of viral integration between patient cells and in vitro infected cells. It would be interesting if the authors could comment on these findings in the Discussion section.

5) The authors convincingly show that only a minor fraction of latently infected cells becomes GFP+ upon treatment with LRAs. Are there GFP- KO2-cells in the culture before and after treatment with LRAs (and if yes, which percentage?).

It would be important to analyze the effect of sequential (e.g., 2) rounds of treatment with LRAs. Another important experiment, which the authors may consider to strengthen the conclusions, is to inhibit H3K27me3 and H3K9me3 followed by treatment with LRAs. Lastly, please clarify why the time of LRAs application is different for in vitro infected versus patient samples (such as 6 hours vs 24 hours).

6) The CD3/CD28 beads (positive control) only reactivated HIV in 3.1% of the latently infected cells. Can the authors comment on why they used 0.5 CD3/CD28 beads/cell for their infection as this is less than the 1 bead per cell recommended by the manufacturer.

Did the authors try other positive controls such as PMA and Ionomycin as reported elsewhere (Laird et al., 2015)? Additionally, can the authors comment on why the latently infected cells were not incubated with the stimuli for longer than 24 hours? This might have resulted in a different outcome.

7) The authors compare the activation status of non-infected, reactivated and non-reactivated cells and conclude that "Overall, the comparison of the latent populations showed very little difference". Judging from the data, the conclusion appears too strong to me. There is a significantly lower percentage of CD25+CD69+ in non-reactivated cells which leaves open the possibility that, at least in part, this population has an incomplete/lower activation (which might partially account for lack of reactivation). The addition of a late marker of activation such as HLA-DR would also greatly strengthen the conclusions.

An important control would be to repeat this experiment with another reactivating stimulation (e.g. PHA) and see if these differences persist. If so, the possibility that lack of full activation accounts for lower viral reactivation must be acknowledged and discussed.

8) The number of integration sites appears to be too low for RLIC (only 162). Please disclose how many genes are hit by these 162 IS. There is no tabular representation of the IS sequencing information. A file with all sorted IS is missing.

Is the value for RLIC really 0% in Figure 5? A significantly higher expression (upon CD3-CD28 activation) of genes containing integration sites of non-reactivated HIV versus reactivated virus seems counterintuitive. As mentioned before the possibility that the lack of complete activation upon CD3-CD28 treatment characterizes the non-reactivated population must be further analyzed and discussed.

9) Clarification on which method was used for the quantification of gene expression is needed. The method presented is not as cited as it requires at least two housekeeping genes. (ddCT method is referenced in the Materials and methods section).

10) In the Discussion section the authors state that "productive proviruses predominantly target actively transcribed regions….On the other hand, non-inducible latent proviruses are observed to integrate into silenced chromatin…". The data in Figure 7A show that the majority of NRLIC viruses still integrate into enhancer and transcriptionally active regions of the genome although integrations in the NRLIC population occur to a higher extent in silenced regions of the genome compared to PIC and RLIC populations. This observation is not addressed in the discussion and the statement is, therefore, a bit misleading.

---

## [Author Response]

Essential revisions:1) Throughout the paper the definition of productively infected cells appears to change. For example, in Figure 1D the ratio of latent to productively infected cells, appears to be calculated by combining csGFP+mKO2- and csGFP+mKO2+ populations whereas in the text corresponding to Figure 1E and 1F the productively infected population is defined as csGFP+mKO2+ cells only. While the csGFP+mKO2- population is reduced in the HIV_GKO_ model, this population is not negligible and should be consistently included when examining the productively infected population.

We agree with the reviewer and have edited the text corresponding to Figure 1E and 1F according to the reviewers’ suggestion. The productively infected cells indeed comprise both csGFP+mKO2- and csGFP+mKO2+ populations. GFP+ mKO2- cells are now included in productively infected cells in all experiments.

2) As the authors aim to establish a superior HIV latency model they should directly quantify the percent increase in latently infected cells compared to their previous DuoFluo reporter instead of only stating that they can identify a 'much larger' number of latent cells.

We have addressed this concern in the new version of the manuscript and added a supplemental figure (Figure 1—figure supplement 1A–C). The data shows that increasing the input of HIV_GKO_ proportionately increased latently and productively infected cells (Figure 1—figure supplement 1A and 1B) with no change in the ratio of latent/productive infection (Figure 1—figure supplement 1C). This was not observed with our previous DuoFluo reporter (Figure 1—figure supplement 1A–C).

3) Throughout the paper the authors express surprise at the low level of reactivation of the latent reservoir observed while also stating that this observation is in agreement with the current literature giving a confusing impression of the novelty of their results. Please amend.

The novelty of the results is our ability to quantify the fraction of latently infected cells that reactivate using our novel dual fluorescent virus. These results are in agreement with what has been observed in latently affected cells isolated from patients and reactivated in vitro. We have edited our manuscript to clarify this point.

4) The fact that JQ1 and bryostatin induce reactivation in patient samples but only little effect on in vitro infected cells is a very interesting finding, pointing to differences in the sites of viral integration between patient cells and in vitro infected cells. It would be interesting if the authors could comment on these findings in the Discussion section.

We are somewhat confused by this question since JQ1 and bryostatin respectively induce 1.7- and 6.8-fold increase of HIV mRNA in patients’ samples, and 1.1- and 6.2-fold increase in HIV_GKO_ latently infected cells. These reactivation levels appear very similar. We are not sure what the reviewers want us to comment on.

5) The authors convincingly show that only a minor fraction of latently infected cells becomes GFP+ upon treatment with LRAs. Are there GFP- KO2-cells in the culture before and after treatment with LRAs (and if yes, which percentage?).

There were indeed csGFP-, mKO2- cells before and after LRA treatment in the case of Figure 2B (RT-qPCR quantification of HIV latency reactivation) (about 97% of csGFP-, mKO2-, and 3% of GFP-, mKO2+ after sort). In the experiment shown in Figure 3 (Flow cytometry of HIV latency reactivation), there were no csGFP-, mKO2- cells after the sort. We isolated pure latent cells (GFP-, mKO2+), and proceeded to LRA reactivation on this pure population.

It would be important to analyze the effect of sequential (e.g., 2) rounds of treatment with LRAs.

We agree with the reviewers since HIV latency reversal has been shown to be stochastic (discussed in the Discussion section). We have tried this experiment several times. Briefly, primary CD4 T cells were infected with HIV_GKO_, and latent cells were isolated by sorting at day 5 post-infection. We, then, isolated again the non-induced HIV_GKO_ latently infected cells. However, these cells showed very low viability and we were unable to restimulate them and answer this question.

Another important experiment, which the authors may consider to strengthen the conclusions, is to inhibit H3K27me3 and H3K9me3 followed by treatment with LRAs.

We thank the reviewers for their suggestion. Several teams have already performed this experiment using different models, and they all reached the same conclusion: the inhibition of H3K27me3 sensitizes latent provirus to LRAs (Friedman et al., 2011; Nguyen et al., 2017; Tripathy et al., 2015).

EZH2 is a key component of the Polycomb Repressive Complex 2 (PRC2) silencing machinery, and the enzyme is required for H3K27me3 deposition. In their study, Friedman *et al.* found that knocking down of EZH2 in latent Jurkat T-cells induced up to 40% of the latent HIV proviruses (Friedman et al., 2011). Knockdown of EZH2 also sensitized latent proviruses to external stimuli, such as T-cell receptor stimulation, and slowed the reversion of reactivated proviruses to latency. Similarly, cell populations that responded poorly to external stimuli carried HIV proviruses that were enriched in H3K27me3. Similarly, but with a different model, Tripathy et al. show that GSK343, a potent and selective EZH2/EZH1 inhibitor, reduced H3K27me3 of the HIV provirus in primary resting cells (Tripathy et al., 2015). This epigenetic change was not associated with increased proviral expression in latently infected resting cells. However, following the reduction in H3K27me3 at the HIV LTR, subsequent exposure to the HDACi vorinostat resulted in increases in HIV gag RNA and HIV p24 antigen production that were up to 2.5-fold greater than those induced by vorinostat alone. Finally, a recent study from Nguyen *et al.* found that the pharmaceutical inhibition of EZH2 in the Th17 primary cell model of HIV latency or resting memory T cells isolated from HIV-1-infected patients receiving highly active antiretroviral therapy, was sufficient to induce the reactivation of latent proviruses (Nguyen et al., 2017). The methyltransferase inhibitors showed synergy with interleukin-15 and VOR.

EHMT2 and SUV39H1 are respectively components of the H3K9me2 and H3K9me3 silencing machinery. In their study, Friedman et al. show that SUV39H1 can contribute to HIV-1 latency in Jurkat T cells, but it appears to be less effective than EZH2 (Friedman et al., 2011). Nguyen et al. found that the inhibition of H3K9me2 in the Th17 primary cell model of HIV latency or resting memory T cells isolated from HIV-1-infected patients receiving highly active antiretroviral therapy, was sufficient to induce the reactivation of latent proviruses (Nguyen et al., 2017).

In conclusion, data showing that inhibiting H3K27me3 and H3K27me2/3 sensitizes latent provirus to downstream LRAs treatment is available and supports our model. We have added text in our Discussion section describing these studies.

Lastly, please clarify why the time of LRAs application is different for in vitro infected versus patient samples (such as 6 hours vs 24 hours).

1) Experiments with patients’ samples and in vitro latently infected cells are different. In the case of in vitro experiment, we are sorting cells after infection, while we are not sorting cells in the case of patients’ samples.

2) In preliminary experiments, we noticed that sorted cells showed a spontaneous reactivation of latent HIV which became more prominent after 24 hours and was minimum at 6 hours (Author response image 1). We therefore focused on analyzing the cells 6 hours after sorting and addition of LRA to eliminate this confounding variable.

**Author response image 1. respfig1:** Time-course plot of percent of GFP+ cells (bars) and live cells (lines) after sort of latently infected CD4^+^ T-cells (n = 2, mean + SEM).

Briefly, CD4^+^ T cells were purified from healthy donors’ blood and activated for 72 hours with aCD3/CD28 beads + 100 U/ml IL-2 before infection with HIV_GKO_. Five days post-infection, latent cells (csGFP- mKO2+) cells were sorted, put in culture, and every 6 hours post-sort, aliquots of cells were stained with live/dead marker and fixed before FACS analysis.</Author response image 1 title/legend>

6) The CD3/CD28 beads (positive control) only reactivated HIV in 3.1% of the latently infected cells. Can the authors comment on why they used 0.5 CD3/CD28 beads/cell for their infection as this is less than the 1 bead per cell recommended by the manufacturer.

In our preliminary experiments, we did not observe any difference between 0.5 bead/cell and 1 bead/cell when infecting cells (data not shown). Given the high price of these beads, we decided to use 0.5 bead/cell to activate cells before infection. However, when we reactivated latently infected cells, we used 1 bead/cell (specified in the Materials and methods section).

Did the authors try other positive controls such as PMA and Ionomycin as reported elsewhere (Laird et al., 2015)?

In preliminary experiment, we indeed used PMA+Ionomycin, but saw similar reactivation as aCD3/CD28 in patients’ samples and HIV_GKO_ latently infected cells after 24 hours of treatment (Author response image 2).

**Author response image 2. respfig2:** (1)Intracellular HIV-1 mRNA levels in rCD4s, obtained from infected individuals and treated ex vivo with a single LRA or a combination of two LRAs for 24 hours, presented as fold induction relative to DMSO control. (n = 4, mean + SEM).

(2) Intracellular HIV-1 mRNA levels in HIV_GKO_ latently infected CD4^+^ T-cells, and treated with a single LRA or a combination of two LRAs for 24 hours in presence of raltegravir, presented as fold induction relative to DMSO control.

(n = 3 (different donors), mean + SEM, paired t-test).

Experiments were performed as for Figure 2 in the main manuscript. </Author response image 2 title/legend>

Additionally, can the authors comment on why the latently infected cells were not incubated with the stimuli for longer than 24 hours? This might have resulted in a different outcome.

We did not treat latent cells with stimuli for longer than 24 hours since most of the literature is using 24 hours. However, we looked at latency reversal at 24, 36 and 48 hours post-treatment, and did not see much differences. We actually did observe some decrease of latency reversal after 48 hours of treatment with panobinostat, likely due to drug toxicity (Author response image 3).

**Author response image 3. respfig3:** Percentage of GFP+ cells are shown after stimulation of latently infected CD4^+^ T-cells with LRAs for 24-, 36- or 48 hours (n = 2 (different donors), mean + SEM, paired t-test).

Briefly, CD4^+^ T cells were purified from blood of healthy donors and activated for 72 hours with aCD3/CD28 beads and 100 U/ml IL-2 before infection with HIV_GKO_. Five days post-infection, latently infected cells (csGFP- mKO2+) cells were sorted, put back in culture, and stimulated with different LRAs in presence of raltegravir for 24-, 36- or 48 hours before performing FACS analysis.</Author response image 3 title/legend>

7) The authors compare the activation status of non-infected, reactivated and non-reactivated cells and conclude that "Overall, the comparison of the latent populations showed very little difference". Judging from the data, the conclusion appears too strong to me. There is a significantly lower percentage of CD25+CD69+ in non-reactivated cells which leaves open the possibility that, at least in part, this population has an incomplete/lower activation (which might partially account for lack of reactivation). The addition of a late marker of activation such as HLA DR would also greatly strengthen the conclusions.

We agree with the reviewers’ comment and repeated the experiment with HLA-DR as a marker along with CD69 and CD25 (Figure 4 of the revised manuscript).

However, the data with these 4 donors do not show significant lower percentage of CD25+/CD69+. Furthermore, the data still show that the majority of the phenotype for both RLIC and NRLIC are highly activated cells (around 70% of cells expressing both early and intermediate activation markers CD69 and CD25 respectively). No difference is observed when looking at HLA-DR.

We conclude that there is no reproducible difference in activation markers between reactivated vs. unreactivated cells. These new data are shown in the revised manuscript (Figure 4).

An important control would be to repeat this experiment with another reactivating stimulation (e.g. PHA) and see if these differences persist. If so, the possibility that lack of full activation accounts for lower viral reactivation must be acknowledged and discussed.

In addition of looking at the RLIC and NLRIC populations activation status after aCD3/CD28, we also looked at these populations’ phenotype after PHA treatment (Author response image 4)) as suggested by the reviewers, and at populations with no stimuli (DMSO – Author response image 4)).

Interestingly, PHA treated cells have a very similar profile to untreated (DMSO) cells at 24 hours post treatment, and it seems that PHA is not as effective as aCD3/CD28 in activating cells, at least 24 hours post-treatment.

**Author response image 4. respfig4:** T-cell activation patterns between double negative, reactivated (RLIC) and non-reactivated (NRLIC) latently infected cells. Briefly, CD4^+^ T-cells were purified from blood of four healthy donors and activated for 72 hours with αCD3/CD28 beads and 20 U/ml IL-2 before infection with HIV_GKO_. At 4 days post-infection, csGFP- were sorted, cultured overnight and stimulated with DMSO (1) or PHA (2). At 24 hours post-treatment, cells were stained for CD25, CD69, and HLA-DR activation markers before performing FACS analysis. (n=4, mean + SEM, paired t-test; *p<0.05; **p<0.01).

8) The number of integration sites appears to be too low for RLIC (only 162). Please disclose how many genes are hit by these 162 IS.

There is indeed a small number of RLIC, since we do have a very small population to work with. As shown in Figure 6A of the manuscript, 80.9% of RLIC IS are within genes (= 131 IS are within genes).

There is no tabular representation of the IS sequencing information. A file with all sorted IS is missing.

We provided an excel file Source data 1 with all IS for each donor and each population. This file comprises 4 tabs. The first tab is the complete list of integration sites for all populations (PIC, RLIC, NRLIC), and the 3 donors. The second tab is the complete list of integration sites for PIC population from the 3 donors, the third tab is the complete list of integration sites for RLIC population from the 3 donors, and the last tab is the complete list of integration sites for NRLIC population from the 3 donors. Each excel tab has 20 columns (Donor, Status, Chromosome, Start, End, Size, Strand, Expression over integration site (same strand), expression over integration site (opposite strand), gene symbol, Ensemble accession number, Description gene, Start gene, End gene, Distance to closest gene, Gene strand, Expression gene, Integration UTR/intron, Direction integration and gene).

Is the value for RLIC really 0% in Figure 5?

Yes, the value is truly 0% in Figure 5.

A significantly higher expression (upon CD3-CD28 activation) of genes containing integration sites of non-reactivated HIV versus reactivated virus seems counterintuitive. As mentioned before the possibility that the lack of complete activation upon CD3-CD28 treatment characterizes the non-reactivated population must be further analyzed and discussed.

We agree with the reviewers and were surprised as well not to observe a correlation between αCD3/CD28-induced latent proviruses integration sites and genes responsive to T-cell activation signals. However, that is what our analysis reveals, and the good activation phenotype of αCD3/CD28 treated cells cannot explain the lack of correlation.

9) Clarification on which method was used for the quantification of gene expression is needed. The method presented is not as cited as it requires at least two housekeeping genes. (ddCT method is referenced in the Materials and methods section).

We edited the text and provide a more detail protocol in the Materials and methods section regarding the normalization of gene expression. Since we didn’t have the standard used in Bullen et al., 2014, we measured relative (instead of absolute HIV mRNA) copy number. We believe that this normalization is appropriate since we are reporting fold changes relative to DMSO. We used the same primers and probe as Bullen et al., 2014, but used a modified qPCR protocol that measures relative copy number (and used GAPDH to normalize the data). We changed the reference for the ddCT method.

10) In the Discussion section the authors state that "productive proviruses predominantly target actively transcribed regions….On the other hand, non-inducible latent proviruses are observed to integrate into silenced chromatin…". The data in Figure 7A show that the majority of NRLIC viruses still integrate into enhancer and transcriptionally active regions of the genome although integrations in the NRLIC population occur to a higher extent in silenced regions of the genome compared to PIC and RLIC populations. This observation is not addressed in the discussion and the statement is, therefore, a bit misleading.

We have edited the text to address this point (Discussion section).